# Deciphering the explanatory potential of blood pressure variables on post-operative length of stay through hierarchical clustering: A retrospective monocentric study

Jérôme Cartailler[1,2], Victor Beaucote[1], Bernard Trillat[3], Etienne Gayat[1,2], Morgan Le Guen[4,5], Alexandre Vallee[6], Marc Fischler[4]*

1 Department of Anesthesiology and Critical Care Medicine, Hôpital Lariboisière, Paris, France, 2 U942 MASCOT, Inserm, Université Paris Cité, Paris, France, 3 Department of Information Systems, Hôpital Foch, Suresnes, France, 4 Department of Anesthesiology, Hôpital Foch, Suresnes, France, 5 Université Paris-Saclay, Gif-sur-Yvette, France, 6 Department of Epidemiology and Public Health, Hôpital Foch, Suresnes, France

* m.fischler@hopital-foch.com

**Data Availability Statement:** All relevant data for the paper is publicly accessible in the Dryad Digital Repository (doi: 10.5061/dryad.12jm63z5r).

## Abstract

### Objective

Mean arterial pressure is widely used as the variable to monitor during anesthesia. But there are many other variables proposed to define intraoperative arterial hypotension. The goal of the present study was to search arterial pressure variables linked with prolonged postoperative length of stay (pLOS).

### Design

Retrospective cohort study of adult patients having received general anesthesia for a scheduled non-cardiac surgical procedure between 15th July 2017 and 31st December 2019.

### Methods

pLOS was defined as a stay longer than the median (main outcome), adjusted for surgery type and duration. 330 arterial pressure variables were analyzed and organized through a clustering approach. An unsupervised hierarchical aggregation method for optimal cluster determination, employing Kendall's tau coefficients and a penalized Bayes information criterion was used. Variables were ranked using the absolute standardized mean distance (aSMD) to measure their effect on pLOS. Finally, after multivariate independence analysis, the number of variables was reduced to three.

### Results

Our study examined 9,516 patients. When LOS is defined as strictly greater than the median, 34% of patients experienced pLOS. Key arterial pressure variables linked with this definition of pLOS included the difference between the highest and lowest pulse pressure values computed throughout the surgery (aSMD[95%CI] = 0.39[0.31–0.40], p<0.001), the

**Funding:** The author(s) received no specific funding for this work.

**Competing interests:** The authors have declared that no competing interests exist.

accumulated time pulse pressure above 61mmHg (aSMD = 0.21[0.17–0.25], p<0.001), and the lowest MAP during surgery (aSMD = 0.20[0.16–0.24], p<0.001).

## Conclusions

By applying a clustering approach, three arterial pressure variables were associated with pLOS. This scalable method can be applied to various dichotomized outcomes.

## Introduction

Intra-operative hypotension (IoH) is widely recognized as a contributing factor to various post-operative complications, including myocardial ischemia, acute kidney injury, and delirium [1, 2]. While numerous definitions of IoH exist [3], a consensus has emerged from the Perioperative Quality Initiative-3 workgroup. They propose a straightforward definition: a mean arterial pressure (MAP) falling below 60–70 mmHg is considered detrimental during non-cardiac surgery [4].

Most studies on IoH primarily focus on specific groups, such as vascular surgery patients, where postoperative complications are detectable through continuous monitoring and frequent biomarker analyses for myocardial ischemia or acute kidney injury. Transitioning from identifying a postoperative complication to measuring the postoperative hospital length of stay (LOS) offers significant benefits, as this outcome is universally accessible and easily obtained. Prolonged LOS (pLOS) may indicate severe postoperative complications, but nearly half of the cases can also be due to nonclinical reasons. Therefore, LOS should be viewed more as an indicator of the hospital process rather than solely as a reflection of comorbidities and care quality. This perspective encourages investigating the complex interplay between IoH, other blood pressure events, and their collective impact on hospital processes [1, 2, 4].

Our study aims to retrospectively analyze a single-center cohort, focusing specifically on the potential statistical relationship between certain intraoperative arterial pressure variables and pLOS. pLOS was defined as a length of stay exceeding median durations, which constitutes our primary objective. Secondary objectives were pLOS based on durations surpassing the 75th and 90th percentiles.

Hence, an automated method for selecting and aggregating relevant arterial pressure variables from non-invasive measurements of mean arterial (MAP), systolic (SAP), diastolic (DAP), and pulse (PP) pressures was introduced. This hypothesis extends to a comprehensive range of potential indicators, such as minimum values, variability, and cumulative time below a given threshold. We propose using a clustering approach followed by effect-size quantification to shed light on the link between intraoperative arterial pressure and pLOS.

## Methods

### Study design, ethics approval and setting

This retrospective study was managed in a tertiary academic private non-profit hospital located in a Paris suburb (France) where surgical activity is multi-purpose, excluding cardiac surgery, and which provides around 20,000 anesthetics a year (including for obstetrics and endoscopy). The study was approved by the local Ethics Committee (Chairperson, Professor Hervé) on the 18th of December 2019 (n˚ 19-11-3). Patients were collectively informed (by means of posters) that their data could be used for research purposes, on condition that the

data were anonymized. This information included the necessary information to enable them to refuse participation. As a result of this procedure, the need for consent was waived by the Ethics Committee. Data were accessed for research purposes from 28th January 2020. Authors had no access to information that could identify individual participants during or after data collection.

## Patient population

Analysis concerned all patients aged 18 years or older who had general anesthesia between 15th July 2017 and 31st December 2019 and stayed in hospital for at least one night. Patients were excluded if operating time was less than 20 min, if they had anesthesia more than once during the same hospitalization, and if they had an obstetric surgical procedure, lung transplantation, interventional radiology, and gastro-intestinal endoscopy and bronchoscopy. Patients with no recorded arterial pressure signal or with aberrant or incomplete signal values were also excluded. All patients were managed according to usual recommendations, especially regarding intraoperative monitoring.

## Data collection

Patient characteristics and preoperative medications were collected from Cesare™, a computerized software for preoperative anesthetic evaluation (Bow Médical, 80440 Boves, France). Centricity Anesthesia software was used to collect intraoperative variables (GE Healthcare, 78 530 Buc, France). LOS and in-hospital mortality were obtained by questioning the health data warehouse.

Arterial pressure measurements were obtained from three time-series: diastolic (DAP), systolic (SAP), and mean arterial pressure (MAP), which were imported from a.csv file. The pulse pressure signal (PP = SAP–DAP) was computed after pairing DAP and SAP using a Cantor pairing function, with time and patient ID as inputs. A Hampel filter, set to five median absolute deviations and constructed over all arterial pressure values, was employed to remove artifact pressure values, which were treated as missing data. Time series were excluded when the interval between two measurements exceeded 10 minutes due to missing values unless these points were at the beginning or end of the recording. The remaining traces were imputed for missing values using a loess smoother with a 40% span parameter. Invasive and non-invasive arterial pressure measurements were combined, we adjusted the continuous blood pressure values to match the sampling frequency typical of non-invasive measurements using interpolation over a 30-second uniformly spaced time vector with a piecewise cubic Hermite interpolating polynomial (pchip).

All relevant data for the paper is publicly accessible in the Dryad Digital Repository (doi: 10.5061/dryad.12jm63z5r).

## Outcomes

Outcome was pLOS defined as the number of days between the date of the surgical procedure and discharge, with a stay strictly longer than median (main outcome), 75th, and 90th percentile (secondary outcomes) used to delineate pLOS. This definition was surgery dependent, indeed to account for variances in pLOS due to surgery complexity and severity. Patients were grouped by surgical class: digestive, thoracic, gynecological, neurosurgical, otorhinolaryngological, urological, and vascular. Each surgery class was further divided into quartiles based on intervention duration, leading to 28 sub-groups. The aim was to reduce the confounding impact of surgery duration, considering its potential correlation with surgery severity and thus pLOS. When pLOS was characterized based on median stay, if the median and third quartile

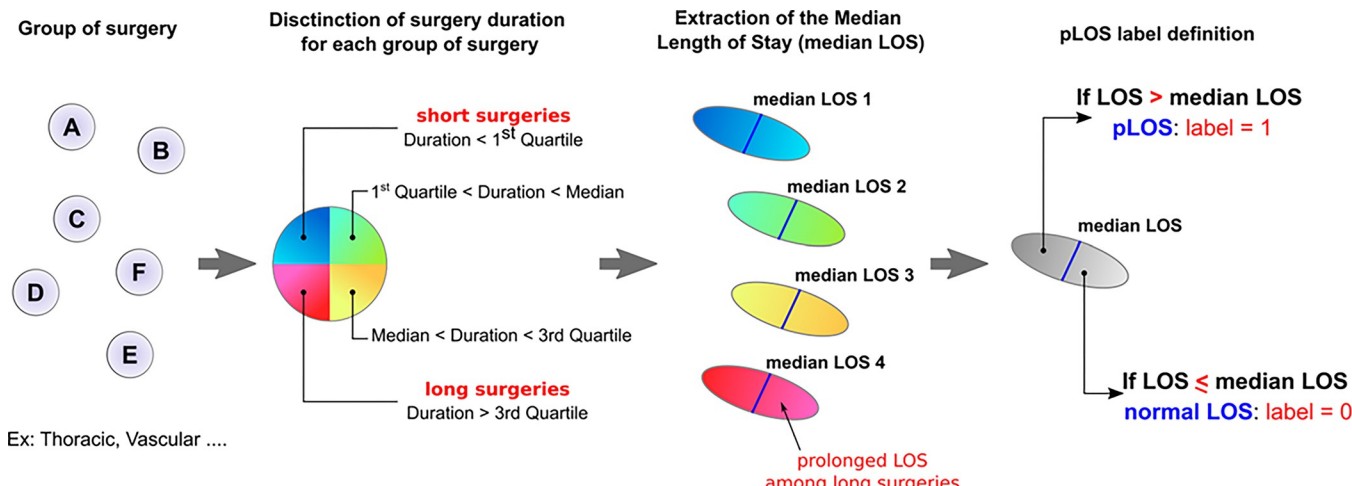

**Fig 1. Construction of prolonged length of stay.** LOS: length of stay. Definition of pLOS outcome starts with seven surgery classes including (pLOS based on median duration): digestive, thoracic, gynecological, neurosurgical, ENT, urological, and vascular interventions. Each class is subdivided into four groups based on surgery duration quartiles. The median LOS and defined pLOS those with LOS > median LOS were computed for each subgroup.

(Q3) were equal, pLOS was defined as a stay longer than the median + one day. This occurs in subclasses where the distribution is tightly clustered. Patients who died post-surgery were categorized as pLOS. Label 1 to pLOS patients and 0 otherwise was assigned (Fig 1).

## Statistical analyses

In analyzing arterial pressure recordings, variables from MAP, SAP, DAP, and PP using a consistent methodology were calculated. This included intra-operative minimum, maximum, mean, median, standard deviation, and variability, with the latter defined as the standard deviation divided by the mean (Fig 2A). 'Drop' variables were defined as the differences between the maximum and minimum values taken throughout the entire anesthesia. For instance, Drop PP refers to the maximal difference of PP observed during the intervention. Additionally, the cumulative time and area below various thresholds were calculated, chosen from every millimeter of mercury between the 5th and 85th percentiles. For each threshold, the cumulative time and area spent below it were scaled and both variables relative to the total intervention time was determined. For instance, CumTimePP>61mmHg designates the duration during the entire surgery the variable PP was higher than 61mmHg. This resulted in 330 features, although the area under the curve was discarded as it was overly redundant with the cumulative time spent and would only add noise and complexity in interpreting the results (S1 Table). The first step was clustering variables and selection of the best candidate within a cluster. The purpose of clustering variables is to group them based on their correlation in an unsupervised manner, independent of the pLOS outcome. A correlation matrix using Kendall's tau for each pair of variables was constructed, employing the native correlation function from the Python v1.4.3 pandas library [5] (Fig 2B). We used Kendall's tau method as it is appropriate for non-parametric distribution and handles ties in ranks. The absolute values of each entry in this matrix were calculated to obtain a measure of the strength of the relationship between variables. This matrix was used to perform a hierarchical agglomeration of the variables using Ward distance. Hierarchical agglomeration groups features into clusters progressively by merging them based on similarity, measured by Kendall's tau. Ward distance is used to minimize variance within each cluster. This agglomeration process can be visualized as a

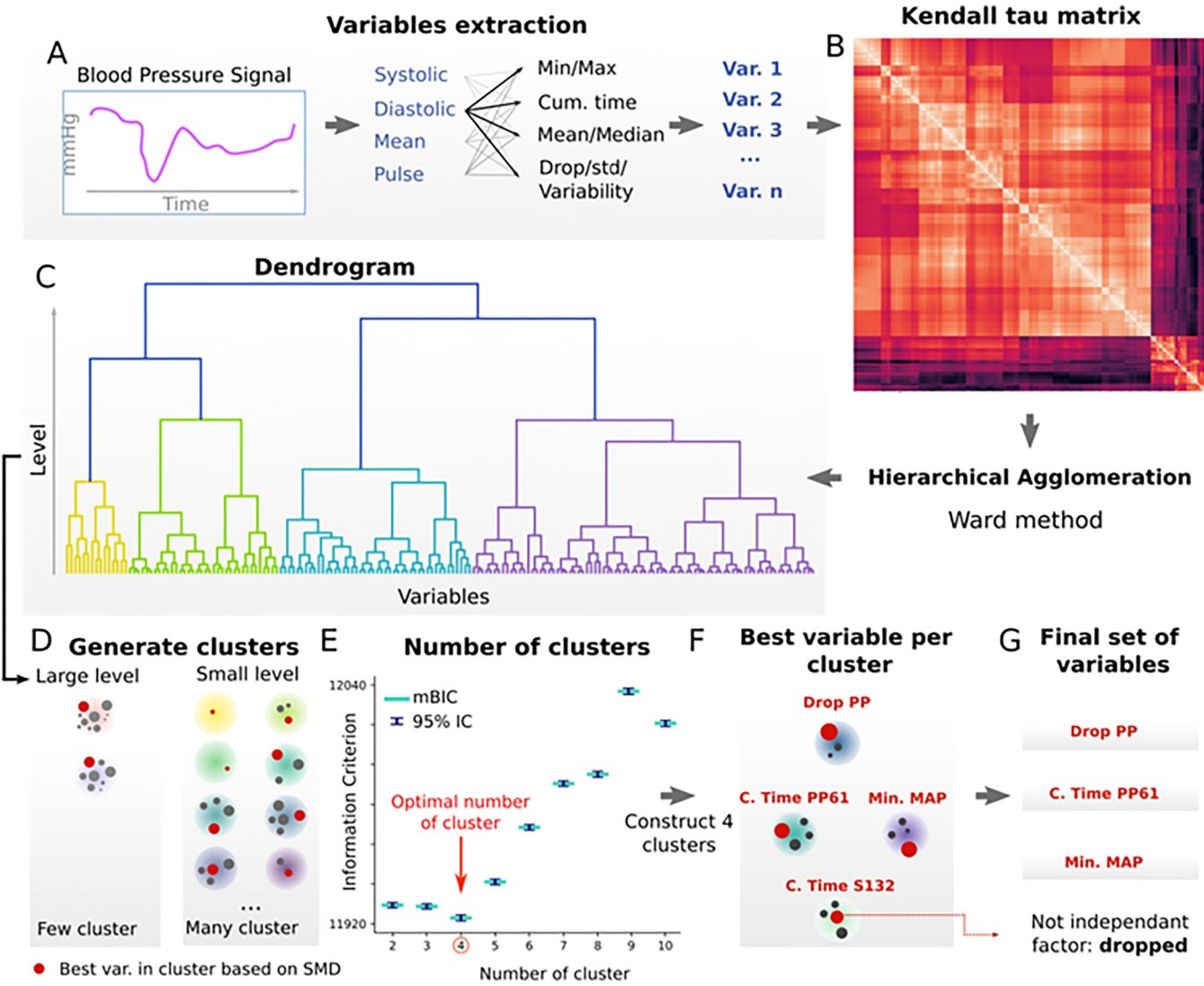

**Fig 2. Variables extraction and selection from arterial pressure signal (pLOS based on median duration).** A: Non-invasive arterial pressure signals include MAP, SAP, DAP and PP that are pre-processed (artefact removal, imputation, interpolation) and from which a panel of variables were computed (ex. Mean, max, cumulative times under given thresholds). B: Dependence between variables was obtained by computing a correlation matrix based on Kendal tau values. C: The matrix was then used for hierarchical agglomeration using Ward method and represented as a dendrogram. D: At this stage, the number of clusters could be arbitrarily chosen modulating a given level of agglomeration. Within each cluster, variables were sorted by aSMD and the best candidate (red dot) were kept. E: To determine the number of clusters to consider, the best variable per cluster was used in a multivariable logistic regression model of pLOS outcome. The number of clusters with the smallest modified Bayesian information criterion mBIC = BIC + (poor p-val) ln(N) were then kept. F: Four clusters represented by the best variables were obtained: smaller pulse pressure values computed over the entire intervention (Drop PP), lowest MAP reached during the intervention (Min MAP), cumulative time PP spent below 61mmHg (CumTimePP>61), and the cumulative time SAP spent below 132mmHg. G: Finally, the independence between variables was tested, resulting in the SAP-related variables being dropped.

dendrogram (Fig 2C). It is important to note that this agglomeration required us to define a level in the hierarchy that would correspond to a certain number of clusters. The linkage function from the Scipy v1.9.0 Python library was utilized for this step [6].

A criterion for selecting the best variable within a given cluster, using effect size, was first established. Within each cluster, variables were ranked according to the absolute standardized mean difference (aSMD), which measures the effect size of a variable between patients with and without pLOS. The aSMD is calculated as $\frac{|\mu_1-\mu_2|}{\sigma_{\text{effect}}}$, where $\sigma_{\text{effect}}$ is the standard deviation in

the effect group, and $\mu_1-\mu_2$ is the mean difference between the two groups [7]. This method is model-free, maintains interpretability compared to methods such as principal component analysis (label-free) or linear discriminant analysis (label-dependent), and unlike ROC AUC, aSMD distributes more linearly and does not require model fine-tuning. Within each cluster, the aSMD was computed for each variable, ranked them, and retained the one with the largest effect size.

Finally, as noted earlier, the number of clusters 'k' is a free parameter. A process to select 'k' using the best variables from different clusters was developed (Fig 2D) and evaluated a likelihood-based performance metric for different 'k' values ranging from 1 to 10 clusters. A given 'k' corresponded to k clusters and thus k best variables. These variables were used as predictors in a multivariable logistic regression model of pLOS. From this model, the Bayesian information criterion (BIC) was calculated, which balances model performance (max log likelihood L) and complexity (number of clusters k), and accounts for sample size (N): BIC = k ln(N) - 2 ln (L) [8]. A penalty equal to ln(N) times the number of insignificant p-values (p≥0.05) in the logistic model was also added to also include variable significance in the model selection process. This penalized BIC thus allows to achieve a balanced model–tradeoff between performance and complexity–while having significant variables. The model, and therefore the number of clusters, with the lowest modified BIC was ultimately selected (Fig 2E and 2F). After identifying the number of clusters using the modified BIC from the logistic model, only the independent variables (with p<0.05) for the final variable selection were retained (Fig 2G). Confidence intervals were estimated from a 1000 iteration bootstrap, and BIC was computed using the statsmodels.api v 0.13.2 Python library [9].

To calculate the sample size, a significance level of 0.05 and a power of 80% (beta = 0.2) was set. Using Hsieh's method [10], a binary outcome (pLOS) with the primary covariate having an odds ratio (OR) of 1.2 was anticipated. A non-informed scenario for prevalence with P0 set at 0.5 was assumed. Given the robust hierarchical interaction among the arterial pressure variables, the sample size for multiple variables was adjusted by applying a correction based on a high squared multiple correlation coefficient 0.95. This adjustment led to a required sample size of 4,740 patients. Note that we took advantage of the larger dataset to refine our machine learning model.

All statistical analyses were conducted using a significant threshold of α<0.05. Ordinal variables were presented as median and interquartile range (IQR), while categorical variables were displayed as count and percentage. Appropriate Fisher or Mann-Whitney tests were used to compare covariates. For logistic models, pLOS was modelled as a random variable from a binomial distribution: pLOS ∼ Binomial (1, ~), where the probability p was derived from the logistic model [11]. To assess a linear change in odds, variables used for odd ratio (OR) estimation were inspected using the R package Hmisc [12]. For instance, a cumulative time with a pulse pressure (PP) above 61mmHg (CumTimePP >61mmHg) exceeding 50 minutes was marked as 1, and 0 otherwise. Drop PP≤25mmHg was set to 25mmHg, 1std = 14.9mmHg. Statistical analyses were performed using R: descriptive statistics and univariate tests, including aSMD, were computed using the CreateTableOne library [13], and forest plots of logistic regression were derived from glm functions.

## Results

Our registry originally consisted of 59,858 patients and 9,516 patients remained for analysis distributed into seven surgical groups: digestive (20.8%), gynecological (13.6%), neurosurgical (21.5%), ENT (10.7%), thoracic (13.7%), urological (15.3%), and vascular (4.5%) (Fig 3).

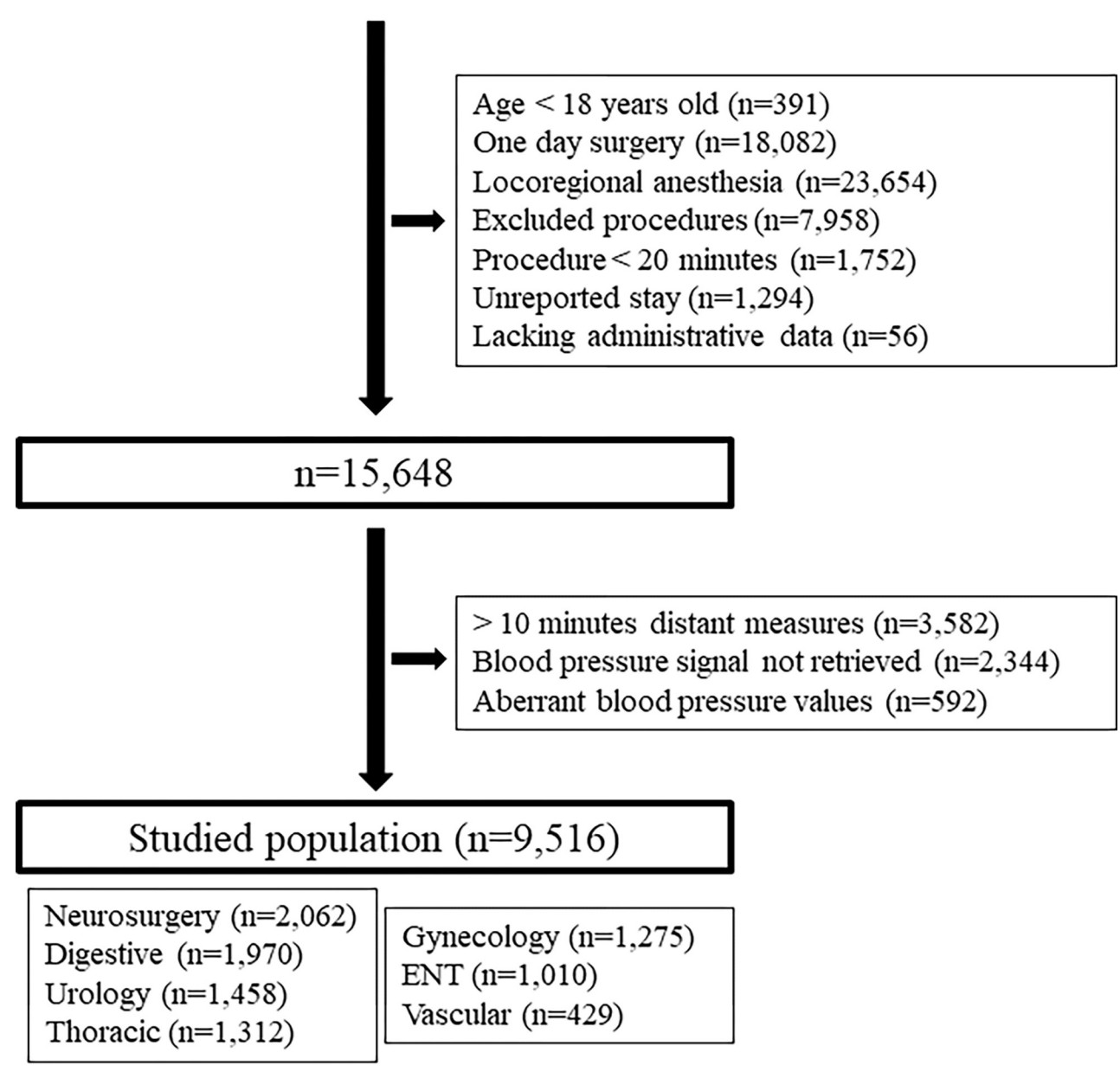

**Fig 3. Flow chart.**

Among these included patients 34% experienced a pLOS, defined as a LOS strictly greater than the median LOS or if they had died post-surgery. The median LOS for all surgical groups was 3 [1–6] days. Clinical characteristics of included patients are reported in Table 1 according to each definition of pLOS.

**Table 1. Patient characteristics for each definition of prolonged length of stay.**

| Groups | All (N = 9516) | pLOS > 50th percentile | | | pLOS > 75th percentile | | | pLOS > 90th percentile | | |
|---|---|---|---|---|---|---|---|---|---|---|
| | | Non-pLOS (N = 6289) | pLOS (N = 3227) | p-value | Non-pLOS (N = 7859) | pLOS (N = 1657) | p-value | Non-pLOS (N = 8684) | pLOS (N = 832) | p-value |
| Age, years | 55.0 ± 17.5 | 53.7 ± 17.3 | 61.2 ± 17.1 | < 0.001 | 55.0 ± 17.5 | 62.5 ± 16.8 | < 0.001 | 55.8 ± 17.5 | 61.1 ± 17.5 | < 0.001 |
| Sex, | | | | < 0.001 | | | < 0.001 | | | < 0.001 |
| female | 4250 (44.7) | 3524 (56.0) | 1464 (45.4) | | 4250 (54.1) | 738 (44.5) | | 4602 (53.0) | 386 (46.4) | |
| male | 5266 (55.3) | 2765 (44.0) | 1763 (54.6) | | 3609 (45.9) | 919 (55.5) | | 4082 (47.0) | 446 (53.6) | |
| BMI, kg/m$^2$ | 25.33 ± 4.7 | 25.4 ± 4.6 | 25.2 ± 4.9 | 0.081 | 25.39 ± 4.7 | 25.02 ± 5.1 | < 0.001 | 25.36 ± 4.7 | 24.93 ± 5.4 | < 0.001 |
| ASA, mean (SD) | 1.93 ± 0.69 | 1.82 ± 0.65 | 2.15 ± 0.72 | < 0.001 | 1.87 ± 0.67 | 2.24 ± 0.74 | < 0.001 | 1.90 ± 0.68 | 2.25 ± 0.78 | < 0.001 |
| 1 | 2584 (27.2) | 1992 (31.7) | 592 (18.4) | < 0.001 | 2317 (29.5) | 267 (16.1) | < 0.001 | 2429 (28.0) | 155 (18.6) | < 0.001 |
| 2 | 5045 (53.0) | 3438 (54.7) | 1607 (49.9) | < 0.001 | 4298 (54.7) | 747 (45.1) | < 0.001 | 4710 (54.3) | 335 (40.3) | < 0.001 |
| 3–4 | 1875 (19.7) | 856 (13.6) | 1019 (31.5) | < 0.001 | 1238 (15.8) | 637 (38.4) | < 0.001 | 1536 (17.7) | 339 (40.7) | < 0.001 |
| Charlson score | 0.64 ± 1.22 | 0.40 ± 0.94 | 1.09 ± 1.52 | < 0.001 | 0.47 ± 1.00 | 1.43 ± 1.72 | < 0.001 | 0.53 ± 1.08 | 1.67 ± 1.89 | < 0.001 |
| **Comorbidities** | | | | | | | | | | |
| Cardio-vascular disease | | | | | | | | | | |
| Hypertension | 13.9 | 12.3 | 21.5 | < 0.001 | 10.8 | 19.9 | < 0.001 | 13.2 | 20.9 | < 0.001 |
| Myocardial ischemia | 2.96 | 2.10 | 7.06 | < 0.001 | 1.62 | 5.58 | < 0.001 | 2.56 | 7.09 | < 0.001 |
| Chronic heart failure | 0.95 | 0.38 | 3.62 | < 0.001 | 0.22 | 2.35 | < 0.001 | 0.54 | 5.17 | < 0.001 |
| Valvular disease | 0.64 | 0.47 | 1.45 | < 0.001 | 0.38 | 1.15 | < 0.001 | 0.58 | 1.32 | 0.019 |
| Arrhythmia | 7.22 | 5.95 | 13.2 | < 0.001 | 4.96 | 11.6 | < 0.001 | 6.67 | 13.0 | < 0.001 |
| Peripheral artery disease | 2.42 | 1.92 | 4.77 | < 0.001 | 1.37 | 4.46 | < 0.001 | 2.11 | 5.53 | < 0.001 |
| Other | | | | | | | | | | |
| Chronic respiratory failure | 9.54 | 7.48 | 19.3 | < 0.001 | 12.3 | 21.5 | < 0.001 | 8.33 | 22.1 | < 0.001 |
| Diabetes mellitus | 7.80 | 6.23 | 15.2 | < 0.001 | 2.10 | 7.06 | < 0.001 | 6.90 | 17.1 | < 0.001 |
| Solid cancer | 7.28 | 5.51 | 15.7 | < 0.001 | 0.38 | 3.62 | < 0.001 | 6.28 | 17.8 | < 0.001 |
| Chronic renal failure | 4.14 | 1.72 | 8.86 | < 0.001 | 2.40 | 12.4 | < 0.001 | 3.09 | 15.1 | < 0.001 |
| Stroke | 0.96 | 0.46 | 1.92 | < 0.001 | 0.52 | 3.02 | < 0.001 | 0.68 | 3.85 | < 0.001 |
| Paresia/Paraplegia | 3.34 | 1.61 | 6.72 | < 0.001 | 2.12 | 9.11 | < 0.001 | 2.65 | 10.6 | < 0.001 |
| Addiction | 3.09 | 2.11 | 4.99 | < 0.001 | 2.53 | 5.73 | < 0.001 | 2.76 | 6.49 | < 0.001 |
| Dysthyroidism | 2.88 | 2.29 | 4.03 | < 0.001 | 2.47 | 4.82 | < 0.001 | 2.58 | 6.00 | < 0.001 |
| Liver Disease | 0.58 | 0.30 | 1.11 | < 0.001 | 0.41 | 1.39 | < 0.001 | 0.46 | 1.80 | < 0.001 |

*(Continued)*

**Table 1.** (Continued)

| Groups | All (N = 9516) | pLOS > 50th percentile | | | pLOS > 75th percentile | | | pLOS > 90th percentile | | |
|---|---|---|---|---|---|---|---|---|---|---|
| | | Non-pLOS (N = 6289) | pLOS (N = 3227) | p-value | Non-pLOS (N = 7859) | pLOS (N = 1657) | p-value | Non-pLOS (N = 8684) | pLOS (N = 832) | p-value |
| HIV | 0.64 | 0.59 | 0.74 | 0.445 | 0.53 | 1.15 | 0.008 | 0.55 | 1.56 | 0.0011 |
| Blood cancer | 0.53 | 0.45 | 0.65 | 0.289 | 0.46 | 0.72 | 0.296 | 0.20 | 0.72 | 0.571 |
| **Surgery** | | | | | | | | | | |
| Digestive | 20.7 | 19.5 | 23.1 | < 0.001 | 20.5 | 21.7 | 0.302 | 20.8 | 19.7 | 0.488 |
| Urological | 15.3 | 14.4 | 17.1 | < 0.001 | 15.0 | 16.8 | 0.076 | 15.4 | 14.7 | 0.616 |
| Thoracic | 13.8 | 13.1 | 15.2 | 0.005 | 13.3 | 15.9 | 0.008 | 14.0 | 11.7 | 0.070 |
| Gynecological | 13.4 | 16.9 | 6.51 | < 0.001 | 14.5 | 8.27 | < 0.001 | 12.9 | 19.0 | < 0.001 |
| Neurosurgical | 21.7 | 19.4 | 26.2 | < 0.001 | 21.1 | 24.4 | 0.003 | 21.8 | 20.7 | 0.493 |
| ENT | 10.6 | 12.6 | 6.66 | < 0.001 | 11.1 | 8.51 | 0.003 | 10.7 | 9.26 | 0.203 |
| Vascular | 4.51 | 4.12 | 5.27 | 0.012 | 4.52 | 4.47 | 0.979 | 4.46 | 5.05 | 0.485 |
| Emergency procedure | 18.0 | 16.8 | 33.7 | < 0.001 | 15.9 | 28.1 | < 0.001 | 17.1 | 27.4 | < 0.001 |
| **Anesthesia** | | | | | | | | | | |
| Total intra-venous anesthesia | 75.1 | 75.1 | 75.0 | 0.942 | 75.3 | 74.0 | 0.270 | 75.3 | 71.9 | 0.028 |

Results are presented as mean ± SD or number (%)

pLOS: prolonged postoperative length of stay; BMI: Body Mass Index; ASA: American Society of Anesthesiologists score; HIV: Human Immunodeficiency Virus; ENT: Ear Nose and Throat surgery

## Arterial pressure variables linked to pLOS based on the median LOS

Through an unsupervised clustering method, the relationship between 330 intra-operative arterial pressure variables and pLOS was analyzed (Fig 2B and 2E). Four key variables emerged as most representative of each cluster: Drop PP (aSMD [95% CI] = 0.39 [0.31; 0.40]), CumTimePP>61 (aSMD [95% CI] = 0.21 [0.17; 0.25]), the cumulative time systolic pressure was below 131mmHg (aSMD [95% CI] = 0.12 [0.08; 0.16], and Min MAP (aSMD [95% CI] = 0.2 [0.16; 0.24]). Each variable showed significant differences between patients with and without pLOS (Table 2). For example, the minimum MAP (Min MAP) was notably lower in patients with pLOS (median [IQR] = 64 [59–70] mmHg versus 62 [57–68] mmHg, p<0.001). Further details on additional variables can be found in S2 Table. After accounting for the four variables in a multivariable logistic regression, cumulative time that systolic pressure stayed below 131mmHg was not significantly linked to pLOS outcomes. The adjusted odds ratios per standard deviation were OR[CI] = 1.29[1.22–1.35] for Drop PP, 0.858[0.817–0.901] for Min MAP, and 1.25[1.10–1.43] for CumTimePP>61.

## Arterial pressure variables linked to pLOS, based on the 75th and 90th percentile of LOS

The same unsupervised clustering method to identify critical arterial pressure variables when pLOS by lengths of stay exceeding the 75th (pLOS75) and 90th (pLOS90) percentiles was used. Of the patients, 1808 (19%) and 832(9%) had a pLOS75 and pLOS90, respectively. For pLOS75, the key associated variables were Drop PP (aSMD = 0.32[0.27, 0.38]) and Min MAP

**Table 2. Univariate comparison of most intra-operative variables derived from arterial pressure (pLOS based on median duration).**

|  |  | No pLoS | pLoS | p-value |
|---|---|---|---|---|
|  |  | n = 6,289 | n = 3,227 |  |
| **Mean arterial pressure** |  |  |  |  |
|  | Min MAP | 64.03 [58.88, 70.32] | 62.38 [57.03, 68.09] | <0.001 |
|  | Max MAP | 107.38 [98.46, 117.06] | 109.39 [99.94, 118.76] | <0.001 |
|  | Drop MAP | 41.97 [32.35, 53.04] | 46.45 [36.46, 56.78] | <0.001 |
|  | Mean MAP | 82.02 [76.27, 88.74] | 82.01 [76.67, 88.43] | 0.514 |
|  | Median MAP | 80.79 [74.46, 88.59] | 80.91 [74.88, 88.22] | 0.595 |
|  | Std MAP | 9.75 [7.50, 12.47] | 10.53 [8.14, 13.20] | <0.001 |
|  | Var MAP | 0.12 [0.09, 0.15] | 0.13 [0.10, 0.16] | <0.001 |
| **Pulse pressure** |  |  |  |  |
|  | Min PP | 33.00 [28.84, 37.67] | 31.74 [27.36, 36.30] | <0.001 |
|  | Max PP | 68.45 [58.35, 81.06] | 74.00 [62.00, 86.63] | <0.001 |
|  | Drop PP | 34.53 [24.25, 47.61] | 41.85 [29.43, 54.06] | <0.001 |
|  | Mean PP | 46.53 [41.82, 52.90] | 47.92 [42.56, 54.72] | <0.001 |
|  | Median PP | 45.21 [40.61, 51.87] | 46.43 [41.27, 53.79] | <0.001 |
|  | Std PP | 7.43 [5.34, 10.43] | 9.04 [6.31, 12.16] | <0.001 |
|  | Var PP | 0.16 [0.12, 0.21] | 0.19 [0.14, 0.24] | <0.001 |
| **Systolic arterial pressure** |  |  |  |  |
|  | Min SAP | 87.03 [81.21, 95.64] | 84.88 [80.03, 92.58] | <0.001 |
|  | Max SAP | 144.42 [132.24, 156.45] | 148.68 [136.92, 159.00] | <0.001 |
|  | Drop SAP | 53.75 [40.90, 67.24] | 60.24 [47.85, 72.14] | <0.001 |
|  | Mean SAP | 110.96 [103.13, 119.51] | 111.47 [104.18, 120.13] | 0.003 |
|  | Median SAP | 109.52 [100.62, 119.57] | 110.05 [101.35, 120.08] | 0.025 |
|  | Std SAP | 12.57 [9.66, 16.20] | 14.09 [10.85, 17.54] | <0.001 |
|  | Var SAP | 0.11 [0.09, 0.14] | 0.13 [0.10, 0.15] | <0.001 |
| **Diastolic arterial pressure** |  |  |  |  |
|  | Min DAP | 51.00 [47.70, 54.52] | 50.53 [46.91, 53.67] | <0.001 |
|  | Max DAP | 82.32 [76.30, 87.20] | 82.24 [76.12, 87.35] | 0.912 |
|  | Drop DAP | 29.94 [24.35, 34.89] | 30.56 [25.28, 35.93] | <0.001 |
|  | Mean DAP | 63.55 [58.75, 68.90] | 62.97 [58.82, 68.25] | 0.024 |
|  | Median DAP | 62.51 [57.20, 69.16] | 61.83 [57.30, 68.30] | 0.043 |
|  | Std DAP | 7.07 [5.56, 8.75] | 7.06 [5.54, 8.81] | 0.856 |
|  | Var DAP | 0.11 [0.09, 0.14] | 0.11 [0.09, 0.14] | 0.373 |

Values are expressed as median [25th percentile– 75th percentile]

MAP: mean arterial pressure; PP: pulse pressure; SAP: systolic arterial pressure; DAP: diastolic arterial pressure

Min: minimal value observed during anaesthesia; Max: maximal value observed during anaesthesia; Drop: difference between max and min taken over anaesthesia; Mean: mean of all values measured during anaesthesia; Median: median of all values measured during anaesthesia; Std: standard deviation of all values measured during anaesthesia; Var: variability of all values measured during anaesthesia

(aSMD = 0.17[0.16, 0.23]). After accounting for the two variables in a multivariable logistic regression, Min MAP and Drop PP were found independent. For pLOS90, associated with the longest LOS, only the Drop PP variable remained significant (aSMD = 0.35[0.27, 0.44]). Despite changes in the pLOS outcome, the best variable for each cluster remained consistent, highlighting the robustness of these variables.

## Discussion

Our study presents a data driven method involving a clustering process and an effect-size based comparison to establish a statistical relationship between a comprehensive set of arterial pressure variables and pLOS. This approach excluded the use of clinical characteristics which are known to be predictors of post-operative morbidity and mortality. By analyzing a monocentric dataset of 9,516 patients, larger pulse pressure fluctuations (Drop PP), lower minimum MAP (Min MAP), and a shorter time spent above 61mmHg of PP (CumTime PP>61mmHg) were independently associated with increased pLOS risk. The analysis distilled four clusters from 330 variables, yielding three independent variables. Interestingly, Drop PP remained associated with the 75th and 90th longest LOS. The suggested approach allowed us to analyze a group of variables that are highly interrelated, grouping them into clusters that represent families of related variables. This could facilitate a deeper understanding of the connections between various studies using different arterial pressure variables, thereby enhancing the comparability and interpretation of their findings.

Vernooij and colleagues conducted systematic research, exploring a variety of definitions for IoH [14]. Although each definition had its own relevance, the sheer number made it challenging to establish a unique, clear IoH definition. Additionally, selecting an IoH definition often intertwines with the specific clinical outcome under investigation, such as acute kidney injury or pLOS [3]. With numerous potential factors contributing to IoH, creating a comprehensive overview becomes a considerable challenge.

This study used hierarchical clustering: an unsupervised algorithm independent of the pLOS outcome. Unlike the conventional application of this method to patient data, we focused on variables, grouping them based on observed correlation. This approach enabled us to examine closely related variables such as the cumulative time MAP remained below 65mmHg and 70mmHg (S2 Table). Our analysis could help understanding why low systolic and mean arterial pressure variables, which both belong to cluster 3, have been reported to have similar impacts on clinical outcomes like acute kidney injury, stroke, or myocardial injury [2]. In contrast, cluster 1, which reflects variability in the arterial pressure, likely bears distinct characteristics from cluster 3, find echoes in the literature, for instance with findings from Hirsch et al., who identified arterial pressure fluctuation, rather than hypotension, as a risk factor for postoperative delirium [1]. In essence, we suggest that opting for the most representative variable from each cluster, rather than amalgamating them, could more accurately unveil the effect of an arterial pressure variable group on a specific outcome.

Our method uniquely identifies key predictors within a candidate pool by capitalizing on the data's hierarchical nature and high collinearity. This approach differs from logistic regression models, which often struggle with collinear variables [2, 3] and can lead to a skewed understanding of their roles [15]. By focusing on clusters, we broaden our understanding and avoid arbitrary variable selection. It also stands apart from techniques like LASSO or Elastic Net, which may arbitrarily select 'better' candidates amidst high collinearity [16]. In addition, absolute aSMD was utilized for single arterial pressure variable selection within each cluster. This step, influenced by clinical outcomes, allowed effective effect size ranking, reflecting effect strength, unlike p-values from two-sample tests that solely reject a null hypothesis [17]. Despite aSMD's inability to account for nonlinearities, it was preferred over metrics like ROC AUC, given its advantages in speed, simplicity, and being model-free, avoiding issues such as model selection and hyper-parameter tuning [15, 18].

### Strengths and limitations

This study has notable strengths. Firstly, the large patient cohort allowed for the inclusion of various surgical types, enhancing the robustness of the analysis. The inclusion and non-

inclusion criteria used define a fairly homogeneous patient population. Another strength is that the outcome, pLOS, is easy to find and has no missing data. The proposed method offers a dual perspective: it not only identifies key variables but also uncovers families of variables, potentially bridging diverse research efforts focusing on similar clinical outcomes but different biomarkers. Focusing exclusively on blood pressure variables, this approach reveals clusters linked with pLOS. However, further studies should address the impact of confounding factors on pLOS. Interestingly, it is plausible that important features such as age or comorbidities, like hypertension, could be encompassed within certain clusters identified by this method, a work beyond the scope of this study.

However, this study has some limitations.

Its monocentric nature raises the potential for variations in arterial pressure distribution and pLOS definitions across different centers and countries due to distinct healthcare practices. For example, one might expect longer pLOS in France as compared to Anglo-American countries [19]. Indeed, it is important to note that, in France, the joint responsibility of anesthesiologists and surgeons for postoperative patient management may impact the study's context. LOS represents an outcome that despite being easy to measure can potentially be influenced by factors that cannot be controlled in practice, including non-clinical factors or individual doctor policies. To universalize our findings, future research should quantify the effect variability across multiple centers and validate identified pLOS variables in different international contexts.

In constructing variables, absolute over relative BP values were favored due to ambiguity in defining baseline BP. Hence, our study does not explore the pLOS and relative BP values relationship, a widely considered factor in preventing intraoperative hypotension [4].

Our analysis primarily relies on non-invasive brachial cuff measurements, yielding poor temporal resolution, which could impact the accuracy of variables tied to arterial pressure variability, such as DropPP. While invasive catheter monitoring offers superior temporal resolution, its application is not in line with current medical practices and is reserved for specific interventions. Future research could explore digital photoplethysmography for improved temporal resolution, though its accuracy needs validation [20].

While this study statistically correlates three arterial pressure variables with pLOS, it cannot ascertain whether enhancing these factors will shorten the stay. To affirm the role of these variables, a prospective randomized clinical trial is needed.

## Conclusion

In conclusion, we developed an approach identifying that intraoperative variability in PP, minimum MAP values, and PP duration above 61mmHg are associated with pLOS. Our cluster-based approach made it possible to handle a diverse set of collinear arterial pressure variables, including multiple IoH definitions. This technique reveals clusters of variables that could have similar clinical implications, as well as those that are likely independent. This scalable method could readily be applied to any dichotomized outcomes, such as mortality, acute kidney injury, or myocardial injury. We encourage other researchers to explore this evaluation method in various clinical settings, surgical groups, and patient populations (e.g., elderly, patients with hypertension). Future studies should aim to investigate additional variables and potentially assess the long-term impact of intraoperative management strategies based on these results.

## Supporting information

**S1 Table. Construction of variables from arterial pressure.**
(DOCX)

**S2 Table. Absolute standardized mean distance per cluster.**
(DOCX)

## Acknowledgments

The authors would like to thank Pauline Touche for her administrative help and Polly Gobin for providing English editing.

## Author Contributions

**Conceptualization:** Jérôme Cartailler, Victor Beaucote, Bernard Trillat, Etienne Gayat, Morgan Le Guen, Alexandre Vallee, Marc Fischler.

**Data curation:** Jérôme Cartailler.

**Formal analysis:** Jérôme Cartailler, Bernard Trillat, Etienne Gayat, Morgan Le Guen, Alexandre Vallee.

**Funding acquisition:** Morgan Le Guen, Marc Fischler.

**Investigation:** Jérôme Cartailler, Victor Beaucote.

**Methodology:** Jérôme Cartailler, Bernard Trillat, Alexandre Vallee.

**Project administration:** Jérôme Cartailler, Marc Fischler.

**Software:** Jérôme Cartailler, Victor Beaucote.

**Supervision:** Marc Fischler.

**Visualization:** Jérôme Cartailler, Victor Beaucote, Marc Fischler.

**Writing – original draft:** Jérôme Cartailler, Victor Beaucote, Morgan Le Guen, Marc Fischler.

**Writing – review & editing:** Jérôme Cartailler, Victor Beaucote, Bernard Trillat, Etienne Gayat, Morgan Le Guen, Alexandre Vallee, Marc Fischler.

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
