## [Decision Letter · Decision Letter 0]

2 Jul 2024

PONE-D-24-14441Deciphering the explanatory potential of blood pressure variables on post-operative length of stay through hierarchical clustering: A retrospective monocentric studyPLOS ONE

Dear Dr. Fischler,

Thank you for submitting your manuscript to PLOS ONE. After careful consideration, we feel that it has merit but does not fully meet PLOS ONE’s publication criteria as it currently stands. Therefore, we invite you to submit a revised version of the manuscript that addresses the points raised during the review process.

We look forward to receiving your revised manuscript.

Kind regards,

Eyüp Serhat Çalık

Academic Editor

PLOS ONE

3. Thank you for uploading your study's underlying data set. Unfortunately, the repository you have noted in your Data Availability statement does not qualify as an acceptable data repository according to PLOS's standards.

Additional Editor Comments:

I congratulate the authors for this important study. With a large cohort of patients, this study provides important results on the relationship between intraoperative blood pressure data and length of hospitalization. The manuscript was reviewed by two reviewers and their recommendations are as follows. We look forward to your point-by-point responses to their suggestions and to your re-uploading the revised version of the manuscript after any necessary edits. We wish you success.

Reviewers' comments:

Reviewer's Responses to Questions

**Comments to the Author**

1. Is the manuscript technically sound, and do the data support the conclusions?

Reviewer #1: Yes

Reviewer #2: Yes

2. Has the statistical analysis been performed appropriately and rigorously? 

Reviewer #1: Yes

Reviewer #2: Yes

3. Have the authors made all data underlying the findings in their manuscript fully available?

Reviewer #1: Yes

Reviewer #2: Yes

4. Is the manuscript presented in an intelligible fashion and written in standard English?

Reviewer #1: Yes

Reviewer #2: Yes

5. Review Comments to the Author

Reviewer #1: 1. Technical Soundness and Data Support

Study Design and Methodology:

The manuscript describes a retrospective cohort study involving 9,516 patients who underwent non-cardiac surgery under general anesthesia. The primary aim was to identify intraoperative arterial pressure variables associated with prolonged postoperative length of stay (pLOS) using a hierarchical clustering approach. The study design is appropriate for the research question, and the methodology is clearly outlined.

Data Analysis:

Statistical Methods: The use of Kendall’s tau coefficients and a penalized Bayes information criterion for cluster determination is appropriate. The application of absolute standardized mean distance (aSMD) for variable ranking and multivariate independence analysis ensures that the analysis is rigorous.

Significance Testing: The p-values reported (<0.001) indicate strong statistical significance, supporting the robustness of the findings. However, the manuscript would benefit from a more detailed explanation of the choice of these statistical methods and their assumptions.

Data Support:

The data supporting the conclusions are comprehensive. The availability of the dataset in the Dryad repository (doi:10.5061/dryad.12jm63z5r) complies with the journal's data policy, enhancing transparency and reproducibility.

2. Presentation and Language

Clarity and Structure:

The manuscript is generally well-organized and easy to follow. Each section flows logically from the previous one, with clear headings and subheadings guiding the reader through the study.

Language and Grammar:

While the manuscript is mostly written in standard English, there are a few typographical and grammatical errors that need correction:

In the Abstract: "received general for a scheduled non cardiac surgical procedure" should be "received general anesthesia for a scheduled non-cardiac surgical procedure."

In the Introduction: "encourages to investigate" should be "encourages investigating."

In the Methods section: "this procedure (collective information) meant that need for individual consent" should be "this procedure (collective information) meant that individual consent was not needed."

Intelligibility:

The manuscript is intelligible, with technical terms appropriately defined and explained. However, simplifying some complex statistical descriptions for a broader audience could enhance readability.

3. Results and Interpretation

Results Presentation:

The results are clearly presented, with appropriate use of tables and figures to illustrate key findings. The hierarchical clustering results are well-detailed, showing the association between arterial pressure variables and pLOS.

Interpretation of Findings:

The conclusions drawn from the data are appropriate and supported by the findings. The discussion appropriately contextualizes the results within the existing literature and outlines the clinical implications of the findings.

4. Ethical Considerations

Ethics Approval:

The study received ethical approval from the local Ethics Committee, which is clearly stated in the manuscript. The procedures for anonymizing patient data and providing collective information to patients comply with ethical research standards.

Data Sharing:

The authors have made the underlying data publicly available through the Dryad repository, complying with PLOS ONE’s data sharing policy. This transparency allows for independent validation of the study’s findings.

5. Strengths and Limitations

Strengths:

The large sample size enhances the robustness and generalizability of the findings.

The innovative use of hierarchical clustering provides a detailed understanding of the relationship between intraoperative arterial pressure variables and pLOS.

The thorough methodological description allows for reproducibility and transparency.

Limitations:

The retrospective design limits the ability to infer causality from the observed associations.

Being a monocentric study, the findings may not be entirely generalizable to other settings or populations.

The manuscript could benefit from a discussion on potential confounding factors and how they were addressed.

6. Overall Assessment

The manuscript is technically sound and contributes valuable insights into the relationship between intraoperative arterial pressure variables and postoperative length of stay. The statistical analysis is rigorous, and the data support the conclusions. The manuscript is generally clear and well-written, though minor errors should be corrected. The findings have significant clinical implications for perioperative management and patient outcomes.

Specific Recommendations

Clarify Statistical Methods: Provide a more detailed explanation of the statistical methods used, including the assumptions underlying Kendall’s tau coefficients and the penalized Bayes information criterion.

Address Potential Confounders: Discuss potential confounding variables and how they were controlled for in the analysis.

Improve Language and Grammar: Correct the identified typographical and grammatical errors to improve readability.

Simplify Technical Descriptions: Simplify some of the more complex statistical descriptions to make the manuscript accessible to a broader audience.

Reviewer #2: Introduction

General

It is well conducted research with novel idea and you have written the paper well.

There are some grammatical errors that need work,

Methods

Study setting: It is better if you add more description regarding the hospital size, anesthesia department annual patient volume, …

Ethics approval: what about the institution head? Mention if s/he is aware of the study and consent was taken from her/him as well.

Exclusion criteria are too much. What is the reason for excluding all this population from the study? As it will likely affect the external validity of the study

Discussion: The conclusion needs to be broaden more and you should not miss a chance to call out other researchers to further explore this new PLOS evaluation method. And also include weakness and strength of the study.

6. PLOS authors have the option to publish the peer review history of their article (what does this mean?). If published, this will include your full peer review and any attached files.

Reviewer #1: **Yes: **Abdullah Abbas Saleh Al-Murad

Reviewer #2: **Yes: **Natan Mulubrhan

---

## [Author Response · Author response to Decision Letter 0]

28 Jul 2024

Rebuttal letter

ONE-D-24-14441

Deciphering the explanatory potential of blood pressure variables on post-operative length of stay through hierarchical clustering: A retrospective monocentric study

We thank the Editor and Reviewers for their comments. 

Please find our responses and amendments in the revised text.

Editor’s comment

Comment 1. Please ensure that your manuscript meets PLOS ONE's style requirements, including those for file naming. The PLOS ONE style templates can be found at

and

Response to Comment 1.

We have verified that the revised manuscript meets PLOS ONE's style requirements.

Comment 2. Please provide additional details regarding participant consent. In the ethics statement in the Methods and online submission information, please ensure that you have specified what type you obtained (for instance, written or verbal, and if verbal, how it was documented and witnessed). If your study included minors, state whether you obtained consent from parents or guardians. If the need for consent was waived by the ethics committee, please include this information.

Response to Comment 2.

We have clarified our clinical activity and participant consent as follows: “This retrospective study was managed in a tertiary academic private non-profit hospital located in a Paris suburb (France) where surgical activity is multi-purpose, excluding cardiac surgery, and which provides around 20,000 anesthetics a year (including for obstetrics and endoscopy). The study was approved by the local Ethics Committee (Chairperson, Professor Hervé) on the 18th of December 2019 (n° 19-11-3). Patients were collectively informed (by means of posters) that their data could be used for research purposes, on condition that the data were anonymized. This information included the necessary information to enable them to refuse participation. As a result of this procedure, the need for consent was waived by the Ethics Committee. Data were accessed for research purposes from 28th January 2020. Authors had no access to information that could identify individual participants during or after data collection.”

We have also modified the “Ethics Statement” field of the submission form to be in line with the modifications brought to the text.

Comment 3. Thank you for uploading your study's underlying data set. Unfortunately, the repository you have noted in your Data Availability statement does not qualify as an acceptable data repository according to PLOS's standards. At this time, please upload the minimal data set necessary to replicate your study's findings to a stable, public repository (such as figshare or Dryad) and provide us with the relevant URLs, DOIs, or accession numbers that may be used to access these data. For a list of recommended repositories and additional information on PLOS standards for data deposition, please see https://journals.plos.org/plosone/s/recommended-repositories.

Response to Comment 3:

Our original version was wrong: “The data are not publicly available since they contain information that could compromise the privacy of research participants. Data supporting reported results are available on the Dryad website open-access repository (doi:10.5061/dryad.12jm63z5r).” We have changed the text as follows: "All relevant data for the paper is publicly accessible in the Dryad Digital Repository (doi: 10.5061/dryad.12jm63z5r).” 

Comment 4. Please review your reference list to ensure that it is complete and correct. If you have cited papers that have been retracted, please include the rationale for doing so in the manuscript text or remove these references and replace them with relevant current references. Any changes to the reference list should be mentioned in the rebuttal letter that accompanies your revised manuscript. If you need to cite a retracted article, indicate the article’s retracted status in the References list and also include a citation and full reference for the retraction notice.

Response to Comment 4.

The reference list has been modified to be in line with these recommendations.

Reviewer #1: 

Comment 1. 

Technical Soundness and Data Support

Study Design and Methodology: Data Analysis, Significance Testing

… However, the manuscript would benefit from a more detailed explanation of the choice of these statistical methods and their assumptions.

Data Support

Response to Comment 1.

We thank the reviewer for these comments. Concerning the motivation of statistical methods, here are the main arguments: 

- The use of Kendall’s tau for analysis is primarily motivated by the fact that several variables could not follow a normal distribution (for instance, cumulative time, which by construction cannot be negative). This led to the use of a non-parametric approach applied to all features for homogeneity. For correlation estimation, non-parametric tests are based on ranks and include Kendall’s tau and Spearman’s Rho. We chose Kendall’s tau as it handles ties in ranks without introducing bias and was therefore more appropriate considering the large number of features. 

- Our penalized Bayes Criterion was used to select the model based on a score constructed from three components. First, it includes the metric of likelihood, which represents the probability that the model accurately captures the reality of the data. Second, a penalty term is applied every time a new variable is included, thus reducing the score as more variables are added. This penalty helps to balance performance and complexity of the model. Finally, a third penalty is introduced to ensure that the variables selected in the model are statistically significant in the multivariate approach. This approach allowed us to achieve a balanced model with significant variables.

We have added a sentence to summarize the statistical description for a broader audience as requested: “The first step …. from the Python v1.4.3 pandas library (Figure 2B). We used Kendall’s tau method as it is appropriate for non-parametric distribution and handles ties in ranks. ...“

We have also added a second sentence “From this model... in the model selection process. This penalized BIC thus allows to achieve a balanced model – tradeoff between performance and complexity – while having significant variables.”

Comment 2. Presentation and Language

Clarity and Structure

Language and Grammar: While the manuscript is mostly written in standard English, there are a few typographical and grammatical errors that need correction:

In the Abstract: "received general for a scheduled non cardiac surgical procedure" should be "received general anesthesia for a scheduled non-cardiac surgical procedure."

In the Introduction: "encourages to investigate" should be "encourages investigating."

In the Methods section: "this procedure (collective information) meant that need for individual consent" should be "this procedure (collective information) meant that individual consent was not needed."

Intelligibility: The manuscript is intelligible, with technical terms appropriately defined and explained. However, simplifying some complex statistical descriptions for a broader audience could enhance readability.

Response to Comment 2.

We have modified our text to follow these comments:

- In the Abstract: we have replaced "received general for a scheduled non cardiac surgical procedure" by "received general anesthesia for a scheduled non-cardiac surgical procedure."

- In the Introduction: we have replaced "encourages to investigate" by "encourages investigating."

- In the Methods section: the sentence "this procedure (collective information) meant that need for individual consent" has been withdrawn in the revised manuscript to clarify participant consent (see response to the Editor).

It is difficult to simplify the description of the statistical technique without compromising its reproducibility. In addition to the two additional sentences proposed in response to Comment 1, we have also added the following explanation that we believe will help readers better appreciate the statistical method: “The absolute values …hierarchical agglomeration of the variables using Ward distance. Hierarchical agglomeration groups features into clusters progressively by merging them based on similarity, measured by Kendall’s tau. Ward distance is used to minimize variance within each cluster.“

Comment 3. Results and Interpretation

Response to Comment 3.

We thank the Reviewer for these comments.

Comment 4. 

Ethical Considerations and Data Sharing findings.

Response to Comment 4.

We thank the Reviewer for these comments.

Comment 5. Strengths and Limitations

The retrospective design limits the ability to infer causality from the observed associations.

Being a monocentric study, the findings may not be entirely generalizable to other settings or populations.

The manuscript could benefit from a discussion on potential confounding factors and how they were addressed.

Response to Comment 5. 

We have now modified in the discussion section the following paragraph intitled "Strengths and limitations”: 

“This study has notable strengths. Firstly, the large patient cohort allowed for the inclusion of various surgical types, enhancing the robustness of the analysis. The inclusion and non-inclusion criteria used define a fairly homogeneous patient population. Another strength is that the outcome, pLOS, is easy to find and has no missing data. The proposed method offers a dual perspective: it not only identifies key variables but also uncovers families of variables, potentially bridging diverse research efforts focusing on similar clinical outcomes but different biomarkers. Focusing exclusively on blood pressure variables, this approach reveals clusters linked with pLOS. However, further studies should address the impact of confounding factors on pLOS. Interestingly, it is plausible that important features such as age or comorbidities, like hypertension, could be encompassed within certain clusters identified by this method, a work beyond the scope of this study. 

However, this study has some limitations. …” 

Comment 6. Overall Assessment

The manuscript is technically sound and contributes valuable insights into the relationship between intraoperative arterial pressure variables and postoperative length of stay. The statistical analysis is rigorous, and the data support the conclusions. The manuscript is generally clear and well-written, though minor errors should be corrected. The findings have significant clinical implications for perioperative management and patient outcomes.

Response to Comment 6.

We thank the Reviewer for these comments.

Specific Recommendations.

Clarify Statistical Methods: Provide a more detailed explanation of the statistical methods used, including the assumptions underlying Kendall’s tau coefficients and the penalized Bayes information criterion.

Address Potential Confounders: Discuss potential confounding variables and how they were controlled for in the analysis.

Improve Language and Grammar: Correct the identified typographical and grammatical errors to improve readability.

Simplify Technical Descriptions: Simplify some of the more complex statistical descriptions to make the manuscript accessible to a broader audience.

Response to Specific Recommendations.

We have included additional material to enhance understanding of the method while preserving reproducibility: 

- “The first step … from the Python v1.4.3 pandas library (Figure 2B). We used Kendall’s tau method as it is appropriate for non-parametric distribution and handles ties in ranks. The absolute values of each entry … Hierarchical agglomeration groups the features into clusters progressively by merging them based on similarity, measured by Kendall’s tau. Ward distance is used to minimize variance within each cluster. …“

- “From this model … This penalized BIC thus allows to achieve a balanced model – tradeoff between performance and complexity – while having significant variables. …”

Reviewer #2:

Comment 1.

Introduction-General

It is well conducted research with novel idea and you have written the paper well.

There are some grammatical errors that need work,

Response to Comment 1.

The text has been reviewed to correct grammatical errors.

Comment 2.

Methods

Study setting: It is better if you add more description regarding the hospital size, anesthesia department annual patient volume, …

Ethics approval: what about the institution head? Mention if s/he is aware of the study and consent was taken from her/him as well.

Exclusion criteria are too much. What is the reason for excluding all this population from the study? As it will likely affect the external validity of the study.

Response to Comment 2.

We have added some items regarding the hospital size, anesthesia department annual patient volume, …: “This retrospective study was managed in a tertiary academic private non-profit hospital located in a Paris suburb (France) where surgical activity is multi-purpose, excluding cardiac surgery, and which provides around 20,000 anesthetics a year (including for obstetrics and endoscopy).”

Regarding the position of the institution head, we would point out that the local Ethics Committee comprises several members directly linked to the hospital head. Otherwise, all the hospital's research protocols are known to the executive team, which includes the head of the Research Department.

Regarding the exclusion criteria, the aim of this study was to investigate which of the various variables derived from intraoperative blood pressure measurement are associated with a poorer outcome (increased length of stay). There are many studies on the relationship between intraoperative blood pressure and poor outcome, and all (or nearly all) focus only on patients undergoing surgery, excluding obstetric procedures, and anesthetic procedures in interventional radiology and endoscopy. We have added two other exclusion criteria: very short operations and lung transplant operations. As a result, we did not follow the Reviewer's opinion that the choice of exclusion criteria affects the external validity of the study. We have added in the section untitled "Strengths and limitations” the following sentence: “… The inclusion and non-inclusion criteria used define a fairly homogeneous patient population. …”

Comment 3.

Discussion: The conclusion needs to be broaden more and you should not miss a chance to call out other researchers to further explore this new PLOS evaluation method. And also include weakness and strength of the study.

Response to Comment 3.

We have changed the conclusion as requested: “In conclusion, we developed an approach identifying that intraoperative variability in PP, minimum MAP values, and PP duration above 61mmHg are associated with pLOS. Our cluster-based approach made it possible to handle a diverse set of collinear arterial pressure variables, including multiple IoH definitions. This technique reveals clusters of variables that could have similar clinical implications, as well as those that are likely independent. This scalable method could readily be applied to any dichotomized outcomes, such as mortality, acute kidney injury, or myocardial injury. We encourage other researchers to explore this evaluation method in various clinical settings, surgical groups, and patient populations (e.g., elderly, patients with hypertension). Future studies should aim to investigate additional variables and potentially assess the long-term impact of intraoperative management strategies based on these results.” 

In addition, we have included the following paragraph intitled "Strengths and limitations”: 

“This study has notable stren

---

## [Editor Report · Decision Letter 1]

30 Jul 2024

Deciphering the explanatory potential of blood pressure variables on post-operative length of stay through hierarchical clustering: A retrospective monocentric study

PONE-D-24-14441R1

Dear Dr. Fischler,

We’re pleased to inform you that your manuscript has been judged scientifically suitable for publication and will be formally accepted for publication once it meets all outstanding technical requirements.

Kind regards,

Eyüp Serhat Çalık

Academic Editor

PLOS ONE
---

## [Editor Report · Acceptance letter]

2 Aug 2024

PONE-D-24-14441R1 

PLOS ONE

Dear Dr. Fischler, 

I'm pleased to inform you that your manuscript has been deemed suitable for publication in PLOS ONE. Congratulations! Your manuscript is now being handed over to our production team.

Kind regards, 

on behalf of

Dr. Eyüp Serhat Çalık 

Academic Editor

PLOS ONE